# Optical Genome Mapping for Comprehensive Assessment of Chromosomal Aberrations and Discovery of New Fusion Genes in Pediatric B-Acute Lymphoblastic Leukemia

**DOI:** 10.3390/cancers15010035

**Published:** 2022-12-21

**Authors:** Huixia Gao, Hanli Xu, Chanjuan Wang, Lei Cui, Xiaotong Huang, Weijing Li, Zhixia Yue, Shuo Tian, Xiaoxi Zhao, Tianlin Xue, Tianyu Xing, Jun Li, Ying Wang, Ruidong Zhang, Zhigang Li, Tianyou Wang

**Affiliations:** 1Hematology Center, Beijing Children’s Hospital, Capital Medical University, Beijing 100045, China; 2National Center for Children’s Health, Beijing 100045, China; 3Beijing Key Laboratory of Pediatric Hematology Oncology, Beijing 100045, China; 4Key Laboratory of Major Diseases in Children, Ministry of Education, Beijing 100045, China; 5National Key Discipline of Pediatrics, Capital Medical University, Beijing 100045, China; 6College of Life Sciences and Bioengineering, School of Physical Science and Engineering, Beijing Jiaotong University, Beijing 100044, China; 7Hematologic Diseases Laboratory, Beijing Pediatric Research Institute, Beijing Children’s Hospital, Capital Medical University, Beijing 100045, China

**Keywords:** optical genome mapping, pediatric B-lineage acute lymphoblastic leukemia, gene fusion, copy number variation, structural variation

## Abstract

**Simple Summary:**

Acute lymphoblastic leukemia (ALL) is characterized by a large number of chromosomal, structural aberrations associated with risk stratification and treatment outcome. However, conventional karyotyping, FISH and PCR have many limitations in detecting chromosomal aberrations. The aim of our study was to assess the potential added value of optical genomic mapping (OGM) for identifying chromosomal aberrations. The chromosomal aberrations of 46 children with B-cell ALL were determined by OGM, and the results of OGM were compared with those of conventional techniques. We found that OGM could detect most clinically significant chromosomal aberrations, and that it has a strong ability to detect complex chromosomal aberrations and refine complex karyotypes. In addition, several novel fusion genes and single-gene mutations, associated with important clinical features, were also identified. Our results show that OGM is highly effective in identifying chromosomal aberrations and has important implications for risk stratification of ALL and the pathogenesis of leukemia.

**Abstract:**

Purpose: To assess the potential added value of Optical Genomic Mapping (OGM) for identifying chromosomal aberrations. Methods: We utilized Optical Genomic Mapping (OGM) to determine chromosomal aberrations in 46 children with B-cell Acute lymphoblastic leukemia ALL (B-ALL) and compared the results of OGM with conventional technologies. Partial detection results were verified by WGS and PCR. Results: OGM showed a good concordance with conventional cytogenetic techniques in identifying the reproducible and pathologically significant genomic SVs. Two new fusion genes (*LMNB1*::*PPP2R2B* and *TMEM272*::*KDM4B*) were identified by OGM and verified by WGS and RT-PCR for the first time. OGM has a greater ability to detect complex chromosomal aberrations, refine complicated karyotypes, and identify more SVs. Several novel fusion genes and single-gene alterations, associated with definite or potential pathologic significance that had not been detected by traditional methods, were also identified. Conclusion: OGM addresses some of the limitations associated with conventional cytogenomic testing. This all-in-one process allows the detection of most major genomic risk markers in one test, which may have important meanings for the development of leukemia pathogenesis and targeted drugs.

## 1. Introduction

Acute lymphoblastic leukemia (ALL) is the most common malignancy in children, of which acute B-lymphoblastic leukemia accounts for about 85% of the patients [1]. Studies have shown that there are numerous structural variations (SVs) in leukemic cells. Many SVs correlate to the drug resistance of leukemic cells, or disease occurrence and progression [2,3]. For example, *KMT2A* (*MLL*) translocations, t(17;19)/*TCF3*::*HLF*, haploidy or low hypodiploidy are high-risk biomarkers, t(9;22)/*BCR*::*ABL1* patients require targeted treatment (imatinib/dasatinib), whereas *iAMP21* patients achieve better outcomes when treated intensively. Clarification of cancer cell behavior and further personalization of treatment require precise identification of SVs.

Nowadays, there are some cytogenetic and molecular methods used in SV detection, with different advantages and limitations. Methods used to detect genetic aberrations vary in their resolutions, which may affect the more precise assignment of the gene being tested [4]. For instance, karyotyping analysis allows for the identification of balanced and unbalanced structural anomalies, limited by the proliferation index of the blast, as well as by the poor quality of the metaphases obtained and the low resolution (about 5 Mbp) [5]. Especially in ALL, the low proliferative index of blasts is the cause of common karyotype failure [5]. FISH has a resolution of about 200 Kbp. However, it is targeted and limited in known regions, and cannot achieve a comprehensive genome-wide detection, especially in ALL. The next-generation sequencing (NGS) technology has greatly enhanced resolution and throughput of detection and the promoted discovery of mutations at single or several base pairs. However, the detection of large SVs, such as inversion, is still difficult due to the short reads sequenced. Due to the various forms of gene variation, the determination of genes involved in known, recurrent karyotype abnormalities, mainly comes from the inference of prior knowledge. For example, t (10;11) (p12; q23) mostly forms the *KMT2A*::*MLLT10* fusion gene, but in rare cases may also be cause *KMT2A*::*NEBL* fusion. Thus, a single diagnostic assay that easily identifies clinically significant SVs is highly desirable.

Optical genomic mapping (OGM) is a technology for high-resolution genome reconstruction from single enzyme-labelled DNA molecules. It can detect balanced and unbalanced translocations, CNVs in a range of few kb up to whole chromosomes (aneuploidies), as well as genomic insertions and inversions [6,7,8]. OGM is based on the imaging of labeled and linearized ultra-high molecular weight (UHMW) DNA. Accurate and precise patterns of labels allow us to de novo assemble the human genome, which is compared to the reference genome map, and extract aberrant molecules from alignments, followed by the generation of local consensus, in order to detect SVs.

Here, we describe a clinical validation study to investigate the genetic aberrations of 46 pediatric B-ALL samples using OGM, karyotyping, FISH, WGS and PCR. By analyzing the detection results of OGM, we demonstrate the feasibility of OGM for the detection of well-established, as well as new putative SVs, in ALL.

## 2. Materials and Methods

### 2.1. Study Design

Forty-six patients with newly diagnosed B-ALL, admitted into Hematology/Oncology Center of Beijing Children’s Hospital, Capital Medical University from June 2019 to June 2020, were enrolled in this study. The children were divided into low-, intermediate- and high-risk groups according to the Chinese Children’s Leukemia Collaborative Group (CCLG)-ALL-2008 regimen [9]. Heparinized bone marrow (BM) samples were subjected to routine genetic diagnostic testing (karyotyping, FISH, and PCR). Clinical data, such as patients’ clinical characteristics, treatment and outcome, were retrieved from medical records. This study has been approved by Institutional Review Board of Beijing Children’s Hospital (IEC-C-008-A08-V.05.1) and all patients signed informed consent.

### 2.2. UHMW DNA Isolation, Quantification and Labeling for Optical Genome Mapping

UHMW gDNA extraction from frozen bone marrow aspirates (BMA) was performed following the manufacturer’s protocols (Bionano Genomics, San Diego, CA, USA). For each sample, 1 mL frozen BMA was used as initial material, and a minimum of 1.5 million white blood cells was used as input to isolate UHMW DNA. Briefly, WBC were centrifuged and lysed by Proteinase K, RNase A in Lysis and Binding Buffer (LBB). DNA was precipitated with isopropanol and bound with a nanobind magnetic disk. Bound UHMW DNA was resuspended in the elution buffer. We used Qubit^TM^ dsDNA BR Assay Kit with a Qubit 3.0 Fluorometer (ThermoFisher Scientific, Waltham, MA, USA) to quantify the gDNA. The gDNA isolation was considered successful when the DNA concentration was equal to or above 36 ng/μL and the coefficient of variation (CV) was <0.3. A total of 750 ng UHMW gDNA was labeled specifically, according to the manufacturer’s guidelines, by using the Bionano Prep Direct Label and Stain (DLS) Protocol. The labeled UHMW gDNA was loaded onto the Saphyr chip for linearization and imaging, and the Saphyr chip was operated at maximum capacity with real-time throughput and quality metrics. According to the manufacturer’s instructions, quality and run parameters included: (1) the total DNA collected ≥150 kb; (2) the map rate (the % of Bionano molecules that align to the reference); (3) the N50 (≥150 kb); (4) the average label density (in labels/100 kb); (5) the positive and negative label variance (indicating the percentage of the labels absent in the reference and the percentage of reference labels absent in the molecules, respectively); (6) the effective coverage of the reference.

### 2.3. Structural Variant Calling and Variant Filtering

Variant calling was executed, enabling SV and CNV detection, with the rare variant pipeline (RVP) included in Bionano Solve (v.3.7). The results were analyzed through two distinct pipelines: a CNV pipeline that allows for the detection of large, unbalanced aberrations, based on normalized molecule coverage, and an SV pipeline that compares the labeling patterns between the constructed sample genome maps and a reference genome map. Reporting and direct visualization of SVs were performed with Bionano Access software v.1.7. In order to assess rare SVs only, we filtered out calls present in an OGM dataset of 180 human control samples provided by Bionano Genomics. The software represents the results from both pipelines in a circos plot, a tool which allows for an easy overview of the detected variants at a glance. Of note, the software calls ‘duplications’ that are smaller than 30 kb ‘insertions’, and ‘inversions’ involving segments of 5 Mb or larger are called ‘intra-chromosomal translocations’.

### 2.4. Comparison of Clinically Significant SVs/CNVs Identified by Conventional Testing

To compare OGM data with standard workflows, we used a visual data presentation consisting of circos plots and individual genome browser views. For data filtering, the variant hg38 DLE-1 SV mask, which blocks difficult-to-map regions and common artifacts, was turned on and the following recommended confidence scores were applied: insertion, 0; deletion, 0; inversion, 0.7; duplication, −1; intra- and inter-translocation, 0.05; and copy number, 0.99 (low stringency, filter set to 0). Per sample, prefiltered data were downloaded as SMAP files for SVs and CNVs separately. These SMAP files were used to determine the number and types of aberrations per sample. With the a priori knowledge that OGM reveals structural complexity undiscernible by karyotyping [10,11], we sought to focus exclusively on SVs and CNVs of potential clinical significance. SVs with a variant allele frequency (VAF) of <10% (equivalent to the presence of SVs in 20% of cell fraction) were considered outside the scope of this study. Of note, ‘Whole genome SV and CNV’ views were only enabled in the latest Bionano Access software version, 1.7, to show SVs and CNVs on different chromosomes (Figure 1).

### 2.5. Confirmation of Additional SVs with Whole-Genome Sequencing

Whole-genome sequencing (WGS) was used to confirm the existence of extra SVs. These were identified by OGM, have with potentially clinical significance and were found via the MGI Tech (DNBSEQ-T7) platform [12]. A MGIEasy FS DNA prep kit (BGI, Beijing, China) was used for WGS library construction according to the manufacturer’s instructions. Paired-end sequencing was performed on a DNBSEQ-T7 sequencing instrument, yielding ~150 bp-sized sequencing reads. A raw data quality check was conducted using FastQC (version 0.11.9). Base quality information was obtained from the FastQC results. Then, the filtered files were mapped to the reference human genome (hg38), and the output BAM files were sorted using samtools sort. The following criteria were used to determine whether SVs, detected independently by OGM and WGS, refer to the same event: (1) deletions, insertions, and duplications, detected by WGS, must overlap with the SV interval defined by optical mapping by at least 50%, and the difference in size predicted by the two methods must be less than 30%; (2) For translocation and inversion, the breakpoint detected by WGS must be within 500 Kb of the breakpoint detected by optical mapping, and the SV direction determined by the two methods must be consistent.

Then, GeneFuse software was used to detect gene fusions directly from the original FastQC files, eliminating the influence of alignment results. GeneFuse was able to visualize the detected fusion with the supported reads and inferred fusion protein structures [13].

### 2.6. Verification of LMNB1::PPP2R2B and TMEM272::KDM4B Putative Fusion Genes

IGVTools (version 2.5.1, http://www.broadinstitute.org/igv, accessed on date 4 January 2019) was used to check the MGI Tech (DNBSEQ-T7) BAM files and extract LMNB1-PPP2R2B exon sequences. Based on the predicted sequence of the fusion mRNA, a plasmid, encoding adjacent sequences of the fusion site, was constructed with GV219, and its digested product with BamHI was used as a positive control. The *LMNB1*-*PPP2R2B* fusion sequence was amplified by cDNA-based PCR in 396 ALL cDNA samples, and agarose gel (concentration: 2%) electrophoresis was performed. The sequences of PCR primers were as follows: F: 5′-AGCTGCTCCTCAAGCTATGC-3′; R: 5′-AAGCTGTGGAAAGTCAGCGA-3′ (product size: 220bp). We verified the amplified products with Sanger sequencing.

*TMEM272*::*KDM4B* fused mRNA was detected by cDNA-based PCR amplification in sample 66. The sequences of PCR primers were as follows: F: 5′-ACAATGCCAGGAGGTCTGGA-3′; R: 5′-AGGATTTGTCAGGTGCCTCC-3′ (product size: 98bp). The expression of *KDM4B* mRNA was quantified by the SYBR Green method, with primers as follows: F: 5′-AAGGCCAAGTTCATCTCCTCCGTC-3′; R: 5′-TGCTCAGTGACAGCCGAGAGCGGA-3′ [14].

### 2.7. Statistics

Associations among categorical values were examined using the Chi-square test or a two-sided Fisher’s exact test. The correlation between SVs and clinical features was analyzed by a Spearman correlation test. Analyses and chart production were performed by using R version 3.4.1. *p* < 0.05 was considered statistically significant.

## 3. Results

### 3.1. Clinical Characteristics of Patients and Technical Characteristics of OGM

#### 3.1.1. Patient Characteristics

This study enrolled 20 male (43.5%) and 26 female (56.5%) patients with median age of 4.17 years (1.4–14.7 years). There were 41 patients (90.7%) with the diagnosis of common-B-ALL and 5 patients (9.3%) with pre-B-ALL. There were 13 cases with the *ETV6*::*RUNX1* fusion gene. *IKZF1* deletion, *BCR*::*ABL* and *TCF3*::*PBX1* fusion genes were detected in 3, 2, and 2 cases, respectively. Three patients carried *KMT2A*::*MLLT3*, *KMT2A*::*AF4* fusion genes, and *iAMP21*, respectively. Based on the CCLG-ALL-2008 protocol, patients were divided into standard- (12 cases), intermediate- (24 cases) and high-risk (10 cases) groups. Two cases in the intermediate-risk group died of acute intracranial hemorrhage and bone marrow suppression with severe infection, respectively.

#### 3.1.2. Raw Data Quality and SV/CNV Callings in OGM

We first evaluated the technical performance of the OGM analysis, which resulted in an average N50 of 260 Kbp (212–351 Kbp), average map rate of 82.6% (60.1–92.0%) and average effective coverage of 420.5X (95.15–667.87X). Thus, all the samples conformed with the requirements of quality control of OGM (Appendix A). In total, we identified 71,534 SVs and 1592 CNVs in 46 leukemia samples (Table 1 and Appendix A).

Per sample, an average of 1555.1 SVs was detected in total, comprising 671.6 insertions, 632.6 deletions, 67.5 inversions, 179.5 duplications, 1.5 intra-chromosomal translocations and 2.3 inter-chromosomal translocations. Of all identified SVs in the total cohort, 2204 were rare (Table 1 and Appendix A). This represented, on average 47.0 per sample, including 11.2 insertions, 28.3 deletions, 0.7 inversion, 3.1 duplications, 2.2 intra-chromosomal translocations and 1.5 inter-chromosomal translocations. Regarding the 1592 CNVs calls (1189 gains and 403 losses), there were 34.5 CNVs per sample including 25.8 (range: 0–131) non-masked gains and 8.7 (range 0–137) non-masked losses (Table 1 and Appendix A). It was noteworthy that the RVP analysis automatically masked regions of the genome with unusually high variance in their relative coverage across control datasets (including centromeric and telomeric regions), assuming that high variance regions may be regions of high CNV occurrence in normal healthy individuals [15].

### 3.2. Concordance between OGM and Conventional Cytogenetic Results

We determined molecular and cytogenetic aberrations in the enrolled patients with conventional technologies, including FISH, PCR, and karyotyping. All 46 cases had FISH/PCR results; 45 cases had G-banded karyotype results. Conventional cytogenetic technologies detected 9 types of fusion genes in 19 samples, *IKZF1* gene deletions in 3 cases, and copy number gains of 5 genes in 15 samples, as shown in Table 2, Table 3 and Table 4.

#### 3.2.1. OGM Reaches 100% True Positive Rate for Known Aberrations, except for Specific Gene Regions

We divided the cytogenetic status, revealed by conventional technologies, into three categories (Table 5): negative karyotype (17 cases), aneuploidies only (18 cases) and trans locations and/or aneuploidies (11 cases). OGM identified most of the clinically relevant genomic abnormalities detected by the conventional technologies (Table 2, Table 3 and Table 4), except for *P2RY8*::*CRLF2* fusion gene resulting from a micro-deletion in three patients (case 48, 51, 76; Table 3 and Table 4). The microdeletion-induced *P2RY8*::*CRLF2* fusion gene occurred in the pseudoautosomal region “PAR” in Xp22.33, which was not contiguously covered by OGM.

#### 3.2.2. Refinement of Abnormal Karyotypes and Resolution of Complex Genome by OGM

In line with the advantage of higher sensitivity and resolution of OGM, our data showed that OGM was able to refine the karyotype (Table 2 and Table 3). In sample 48, besides *iAMP21* revealed by FISH, OGM further detected a chromothripsis of chromosome 21 (Figure 2). Intra-chromosomal amplification of chromosome 21 (*iAMP21*) defines a subgroup of pediatric B-ALL, characterized by multiple structural abnormalities of amplification, inversion and deletion, and which has a poor prognosis with standard therapy [16]. The circus plot illustrated the shattering of chromosome 21, resulting in large-scale intra-chromosomal rearrangements. Furthermore, the overall copy number of chr21 was more than 3, and the copy number in the region from 33.2Mb to 39.2Mb, where *RUNX1* lies, was 6. Array-based comparative genomic hybridization (aCGH) revealed that the gene amplification/deletion patterns of abnormal chromosome 21 were significantly different among patients [17]. Consistent with previous studies [17,18], at the OGM-CNV interface, we also see a classical stepwise rise in copy number around the 33–45 Mb region, followed by a sharp drop off to deletion or normal diploid levels. These patterns were consistent with the classical breakage—fusion—bridge (BFB) cycle of oncogene amplification [19]. At the same time, we also observed some secondary genetic changes in this sample, including −7 (5%) [18], *P2RY8*::*CRLF2* (17%) and *IKZF1* (22%) deletion [20], etc. By comparing copy number profiles from the OGM with those from previous microarray studies, the altered complexity observed by the OGM brings an intuitive visual illustration of the BFB mechanism.

On the other hand, high-resolution OGM uncovered the complexity of structural variations and heterogeneity in breakpoint regions that were difficult to resolve with conventional cytogenetic technologies. In five cases with normal karyotype (case 41, 80, 97, 101, 109), OGM showed large copy number changes, ranging from 18,514 bp to 133,785,261 bp on different chromosomes (Table 2). In case 47, karyotyping showed the additional genetic material of unknown origin at 1q21. OGM revealed that some fragments of chromosome 1 amplified and translocated intra-chromosomally. Thus, OGM refined the karyotype as dup(1)(q21.2q24.3-q24.3q32.3) and der(1)t(1;1)(q41;q43) (Figure 3; Table 2). In summary, OGM provides a more accurate and comprehensive insight into the genomic origin of complex karyotypes.

#### 3.2.3. Identification of Novel Chromosomal Alterations or Gene Fusions by OGM

OGM identified several novel chromosomal alterations or gene fusions (Table 2 and Table 3). In case 66, G-banded karyotyping revealed two distinct translocations—t(11;22) (q24.3; q12.2) and t(13;19)(q14.13;q13.3) (Figure 4C). The former leads to a *FLI1*::*EWSR1* fusion, 15. However, the results of the latter have not been reported. OGM revealed a fusion of *TEME272* on chromosome 13 and *KDM4B* on chromosome 19 in an inverted orientation (Figure 4). Furthermore, OGM detected three-way translocations in samples 83, 103 and 114, which were t(4;7;21) (q21.21; p15.3;q11.2), t (12;16;21) (p13.2;q24.3;q22.12) and t (5;12;21) (q11.2;p11.23; q22.12) (Figure 5). In samples 83 and 103, the above-mentioned inter-chromosomal translocations produced *OSBPO3*::*NRIP1* and *SPG7*::*RUNX1* fusion genes, respectively. As far as we know, the above-mentioned putative fusion genes have not been reported. These findings indicate the presence of cryptic or more complex translocations, which may be under-ascertained with current conventional detection methods. Unfortunately, due to running out of previously clinical samples collected, no raw BMA nor DNA is available for verification now. We will focus on these rare three-way translocations in future studies and further verify their clinical significance. Additionally, the above chromosomal SVs, listed in Appendix A, affect several cellular biological processes (Figure A1, detailed in Appendix B) in pediatric B-ALL.

Taken together, OGM serves as a single-platform assay that can identify different types of chromosomal structure variations, which may be difficult to identify with conventional technologies.

### 3.3. Clinical Values of OGM Detection

#### Difference in SV Numbers among the Three Risk Groups

Accurate risk stratification is of great significance for patients’ treatment and prognosis evaluation. We divided the 46 patients with B-ALL into 3 groups based on accurate MICM and their clinical manifestations. There were 12, 24, and 10 cases in standard-, intermediate-, and high-risk groups, respectively. We found that the average number of different types of SVs were similar in different risk groups (*p* > 0.05). Similarly, there were no significant difference in the mean amount of CNV and aneuploidy among the three risk groups (*p* = 0.484 and 0.263, respectively, Figure 6). Thus, the number of chromosomal aberrations is not fully related to patients’ risk stratification, suggesting the important role of some SVs in leukemogenesis and clinical-biological features.

As there was no significant difference in the number of SVs among different risk groups, we further investigated the correlations between recurrent SVs and common clinical characteristics such as MRD at day 15, 33 and 78, age and white blood cell count at diagnosis of the enrolled patients (Table 6 and Table 7). Regarding gene fusions, we found that *AC141586.1*::*KCTD5*, *ATP10A*::*AC016266.1*, *CALCOCO2*::*SUMO2P17*, and *MIR4435.2HG*::*AC017002.5* were significantly positively correlated with d33 MRD; the former three fusions and *PDCD6IPP1*::*AC138649.1* were related to risk stratification. We also found positive correlations of ETV6-AP000331.1 with d15 MRD, and of *AL034430.1*::*SLX4IP* and *MKKS*::*SLX4IP* with d78 MRD, respectively. Some candidate fusion genes, such as *GRAPL*::*KYNUP3, ARL8B*::*EDEM1* and *GPN3*::*FAM216A*, were related to patients’ age, white blood cell count and the percentage of leukemic cells in peripheral blood at diagnosis (Table 6).

In the aspect of single-gene aberrations, d15 MRD was positively correlated with aberrations in *NF1* and *ERG*. The gene d33MRD was positively correlated with abnormalities in *NF1*, *SH2B3, IKZF1, ERG* and *CREBBP*; d78 MRD was also positively correlated with abnormalities in *KMT2A, CREBBP, BTG1* and *PIK3CA*. Moreover, abnormalities in *CREBBP, ERG, KMT2A* and *SH2B3* were all correlated with risk stratification. Aberrations in *BCR*, *ABL1* and *TCF3* were all positively correlated with the number of leukocytes found at diagnosis (Table 7). In addition, *IKZF1* aberrations were positively correlated with age, and the *IKZF1* deletion site were, the same as 7p12.2(50324504_50399656), detected by OGM in patient maps (sample 48, 58, 101) (Figure 7). Patients are classified as *IKZF1*^plus^ positive if they harbor an *IKZF1* deletion plus a deletion involving *CDKN2A/B*, *PAX5*, or PAR1 (positive for *P2RY8*::*CRLF2*), without a concurrent *ERG* deletion [21]. We detected *IKZF1* deletion in 3 cases, among whom patient #48 met the criteria of *IKZF1*^plus^, and also harbored *iAMP21*. The patient was treated with high-risk regimen, with d15MRD of 3.20 × 10^−2^ and negative MRD at the end of induction. The other two non-*IKZF1*^plus^ patients (Case #101 and #58) had increased copy number of *AML1* and *BCR*::*ABL1* fusion, respectively. Case #101 received high-risk treatment with negative MRD at both day 15 and 33. Case#58 was classified into the intermediate-risk group, and died of acute intracranial hemorrhage during induction. Due to the limited number of samples, we could not obtain clinical characteristics of the *IKZF1*^plus^ cases. Importantly, this study suggests the potential of OGM to detect the genetic aberrations of *IKZF1*^plus^ in an all-in-one process. Its clinical value would be further verified in future studies with a large sample size.

In conclusion, we found that some OGM-detected recurrent fusion genes or single-gene aberrations were correlated with clinical risk stratification indicators. Some of these genes have not been reported, and their clinical significance need to be further verified.

### 3.4. Successful Validation of New SVs with Combination of OGM and NGS

#### 3.4.1. NGS Validation of SVs Detected by OGM

Next, based on the amount of partner genes expressed in bone marrow, their relevance to the pathogenesis of leukemia, and whether the promoter region is preserved, five possible fusion genes were selected for WGS verification: *PSPC1*::*ZMYM2* (deletion), *SH2B3*::*ATXN2*(deletion), *LMNB1*::*PPP2R2B* (deletion), *CWH43*::*TPTE* and *TMEM272*::*KDM4B* (inter-chromosomal translocation). The WGS results confirmed the existence of these five gene fusions. However, further analysis suggested that the *SH2B3*::*ATXN2* and *PSPC1*::*ZMYM2* fusion result from the deletion of a 0.02–0.2 Mb region containing promoter sequences of the fusion partners (Appendix A). Thus, these fusions do not lead to the transcription of fused mRNA. With regard to *CWH43*::*TPTE* fusion, neither of the promoters of the two genes remained in the fusion (*CWH43* lost its promoter and 1-9 exon regions, while *TPTE* lost its promoter and 1–13 exons), and so the *CWH43*::*TPTE* fusion could not produce a fused mRNA and or protein (Appendix A). Regarding the *TMEM272*::*KDM4B* fusion gene, resulting from an inter-chromosomal translocation between chr13 and chr19 in sample 66, WGS that revealed the breakpoints are in intron 2 and intron 1 of *TMEM272* and *KDM4B*, respectively. The RT-PCR result confirmed the existence of *TMEM272*::*KDM4B* fused mRNA (Appendix A). Thus, the entire coding sequences of *KDM4B* were under the control of the *TMEM272* promoter in this rearrangement (Appendix A). The mRNA expression of *KDM4B* in sample 66 was 1.69 times higher than that in other patients with newly diagnosed ALL.

In sample 46, the deletion of about a 20Mb region (Chr5: 126,720,525-146,759,262), containing 5′ sequences of *LMNB1* and *PPP2R2B*, leads to fusion of *LMNB1* and *PPP2R2B* on chromosome 5. The breakpoints are located at *LMNB1* intron 2 and *PPP2R2B* intron 6, respectively (Figure 8D). The fusion retains 5′ regulatory regions of both genes, exon 1–2 of *LMNB1* and exon 1–6 of *PPP2R2B* (Figure 8D). Further RT-PCR result confirmed the existence of *LMNB1*::*PPP2R2B* fused mRNA (Appendix A). However, as the two fused mRNA were predicted to produce truncated *LMNB1* (N terminal 172 amino acids encoded by exon 1 and exon 2) and *PPP2R2B* (N terminal 45 amino acids coded by a range from exon1 to exon 6), respectively, we cannot ascertain the existence of the two truncated proteins due to unavailability of leukemic samples. Therefore, we have validated the existence of the above four gene fusions at the genome DNA level (except for *CWH43*::*TPTE,* which lost their respective promoter regions), and further at the mRNA level, for two of them.

#### 3.4.2. Determination of LMNB1::PPP2R2B Fusion mRNA in Another Cohort of B-ALL Patients

Since *LMNB1* plays an important role in nuclear structure and *PPP2R2B* has phosphatase activity inhibiting oncogenesis, we continued to explore the incidence of *LMNB1*::*PPP2R2B* fusion gene in a new cohort of patients. We determined *LMNB1*::*PPP2R2B* fusion of mRNA in diagnostic bone marrow samples of 396 children with B-ALL (diagnosed from October 2018 through March 2021). *LMNB1*::*PPP2R2B* fused mRNA was finally detected in 1 patient (Appendix A). The incidence of *LMNB1*::*PPP2R2B* fusion is estimated at 0.25%. It is interesting that both the patients with *LMNB1*::*PPP2R2B* fusion carried *ETV6*::*RUNX1* fusion, implying that *LMNB1*::*PPP2R2B* fusion played a role in the pathogenesis of *ETV6*::*RUNX1*-positive leukemia.

Taken together, OGM combined with WGS can play an active role in the identifying new genetic alterations, affecting cellular signaling pathways, and in laying a solid foundation for subsequent research.

## 4. Discussion

In this study, we compared the role of OGM with that of conventional cytogenetic technologies in the genotyping of pediatric ALL. The results showed a good concordance between conventional cytogenetic techniques and OGM in identification of abnormalities of various types (balanced or complex translocation, duplication, deletion, insertion, inversion and aneuploidies). The exception was *P2RY8*::*CRLF2* fusion, involved in the pseudoautosomal region of X/Y chromosomes. Noteworthily, *P2RY8*::*CRLF2* fusion alterations have been repeatedly reported to have high clinical significance and are included in the stratification of patient in clinical trials since they enable treatment by TKI inhibitors [22,23,24,25]. In fact, among the 3 patients (cases #48, #51 and #76) in this study, two were in the high-risk group and one was in the intermediate-risk group, and all had poor treatment responses. Thus, identification of *P2RY8*::*CRLF2* fusion is critical in clinical practice. Though OGM has been shown to have the potential to be a routine tool in hematology malignancies [26,27], some technical limitations remain to be further addressed by anticipated software improvements, especially in PAR regions. However, it is also possible that *P2RY8*::*CRLF2* fusion can be missed by VAF which is lower than 5%. Importantly, our results highlight several advantages of OGM. Firstly, OGM allows for the high-throughput, accurate detection of different types of anomalies. OGM has greater sensitivity and resolution than karyotypes, theoretically allowing the analysis of whole genomes and the identification of the aberrations (insertions between 5–50 kbp, deletions > 7 kbp, invertions > 70 kbp, duplications > 150 kbp and transpositions where the translocated fragments are >70 kbp) compared with FISH and PCR. At the same time, OGM is a non-time-consuming method, with only 3.5 days required from sample preparation to standard analysis output. Due to these advantages, OGM can refine the karyotype with high-resolution and uncover the complexity and heterogeneity of structural variations in breakpoint regions, as we can see in the details in Table 2 and Table 3. These may disrupt/impact genes within breakpoint regions, leading to subtle genotype-phenotype differences [26] which were difficult to solve by conventional cytogenetic technologies. Our study also highlights the ability of OGM to identify common fusion genes and reveal novel structural variants. Almost every study using OGM for leukemia showed that a large number of SV that could not be identified by conventional methods could be detected in every sample. Dozens of new inversions, duplications, and hundreds of new insertions and deletions were identified in each patient [28,29]. These SVs involve many genes involved in cell growth, differentiation, and tumorigenesis. Some SVs are located in the inter-gene region, which may lead to abnormal gene expression regulation through the cis-acting elements affecting gene expression regulation.

Accurate detection of known fusion genes is of great significance for risk stratification, and the discovery of new aberrations may lead to important biological insights and new therapeutic methods [30]. Our results show that OGM can detect almost all clinically significant SVs (except for some specific chromosome regions, such as pseudoautosomal region on X/Y chromosomes) reported by cytogenetic methods, as well as those that cannot be identified by conventional methods, and provide a more accurate and comprehensive genomic SV analysis of complex karyotypes. For example, OGM detected a large number of CNVs in 4 cases (case 41, 97, 101, 109) with normal karyotype reports. Most of these CNVs were involved in increased numbers of chromosomes 4, 6, 10, 14, 17, 18, 21 and changes of some segments of chromosome 9, 21. In addition, OGM identified a monosomy of chromosome 7 in case 101, and we also noticed the same presence in samples 48, 71 and 95. Therefore, we speculate that chromosome 7 may have instability factors. These changes may lead to the reduction of important tumor suppressor genes, the damage of gene structural stability or gene expression regulation, which may play an important role in tumorigenesis.

In our study, OGM detected a variety of possible gene fusion events and SVs affected single genes, and some of the recurrent SVs were related to clinical characteristics and risk stratification, as we presented in the results section (see Table 6 and Table 7 for details). These variants lead to destruction or loss of the involved genes, or the formation of gene fusion. However, due to the limited samples included in our study, we have not ascertained the effects of these gene fusions and SVs on the response to treatment and patient management. In future studies with larger sample size, we would focus on the clinical significance of these gene alterations. Since the translocations and corresponding fusion genes analyzed by OGM are mainly conjectural on the basis of gene structure, whether fusion transcripts are generated remains to be confirmed after further verification. For example, in case 47, OGM showed an amplification of a segment and the formation of a derivative chromosome 1 [dup (1) (q21.2q24.3-q24.3q32.3) (149910330_213101514), der(1)t(1;1) (q41;q43)] (Figure 3; Table 3). Genome mapping indicated the duplication and inversion of the ESRRG gene in this event. It was reported that *ESRRG*, a transcriptional activator, regulated the proliferation of breast cancer cells by directly binding to the response element in the promoter of DNA cytosine 5-methyltransferase 1 (*DMNT1)* [31]. Therefore, the mutation of the *ESRRG* gene found in our study may be involved in the occurrence and development of leukemia as an important driving factor. Therefore, OGM would play an important role in clinical practice by further optimizing risk stratification and prognosis evaluation in ALL.

In this study, OGM identified two unreported novel fusions (*LMNB1*::*PPP2R2B*, *TMEM272*::*KDM4B*) in samples 46 and 66, simultaneously, with two known fusions (*ETV6*::*RUNX1, FLI1*::*ESWR1*), respectively. *TMEM272*::*KDM4B* results from a translocation between chromosomes 13 and 19, which makes the coding region of *KDM4B* under the control of the regulatory elements of *TMEM272* (Appendix A). RT-PCR confirmed the presence of the fusion mRNA. The mRNA expression of *KDM4B* was higher than that in other newly diagnosed ALL, suggesting an overexpression of the *KDM4B* caused by the fusion. A number of studies have shown that *KDM4B* is often overexpressed in breast, colorectal, ovarian, lung, gastric and prostate cancer cells, resulting in H3K9me3 demethylation, subsequent gene expression changes and genomic instability to induce tumors [32,33,34,35,36]. Whether *KDM4B* plays the same role in ALL and other hematological malignancies remains to be further explored. In addition, the existence of *TMEM272*::*KDM4B* fusion should be confirmed in a large number of samples in a future study.

The *LMNB1*::*PPP2R2B* fusion is caused by a large fragment deletion (about 20Mbp) in chromosome 5. Both genes retain their 5′ promoter regions. However, *LMNB1* only retains exon 1 and 2, encoding 172 amino acids, and loses its main domains, while the fused *PPP2R2B* sequence encodes 45 amino acid residues and terminates at a stop codon in the reverse direction. *PPP2R2B* contains 7 repeated WD-40 motifs (protein–protein and protein–DNA interaction sites). Deletion results in the retention of only two WD-40 motifs (149 amino acid residues) and the loss of the kinase domain (295th–298th amino acid), while the fused *LMNB1* sequence encodes 5 amino acid residues and terminates at a stop codon in the reverse direction. Although our study confirmed that the fusion produces two fusion mRNAs, cells may directly recognize and degrade the two truncated proteins, resulting in the haplo-insufficiency of the two genes. *PPP2R2B*, a serine/threonine protein phosphatase, is implicated in the negative control of cell growth and division. Tan et al. reported that *PPP2R2B* inactivation could target *PDK1/MYC* signaling to promote growth and resistance to rapamycin of colorectal cancer cells [37]. Recent studies have reported that *PPP2R2B* is a robust tumor suppressor and plays an important role in anti-tumor immune responses, and that its dysregulation could contribute to the onset and progression of breast cancer [38]. As a nuclear structural protein, *LMNB1* contributes to maintaining nuclear morphology and hematopoietic stem cell function. Down-regulation of *LMNB1* expression causes genomic instability due to defective DNA damage repair. Thus, this fusion possibly leads to down-regulation of the expression of the two proteins, which may be related to leukemogenesis. It is worth noting that 2 out of 396 B-ALL patients carried the fusion, and both were *ETV6*::*RUNX1*-positive, suggesting that *LMNB1*::*PPP2R2B* may be involved in the role of *ETV6*::*RUNX1* in leukemogenesis. The underlying mechanism needs to be further explored.

## 5. Conclusions

In summary, two new fusion genes (*LMNB1*::*PPP2R2B* and *TMEM272*::*KDM4B*) were identified by OGM and verified by WGS and RT-PCR for the first time. OGM addresses some of the limitations associated with conventional cytogenomic testing, as this all-in-one process allows the detection of most major genomic risk markers in one test, which has important meanings for the development of leukemia pathogenesis and targeted drugs.

## Figures and Tables

**Figure 1 cancers-15-00035-f001:**
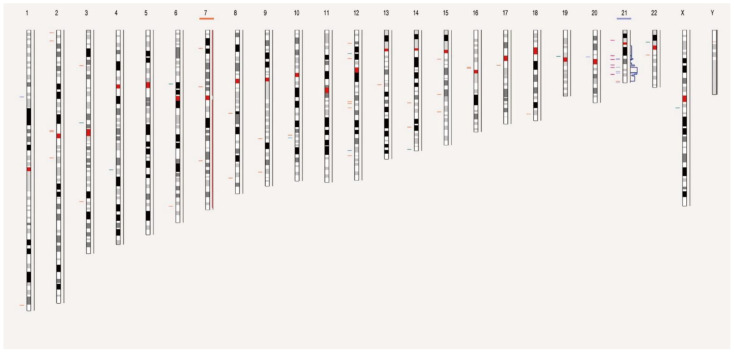
View of the distribution of SVs and CNVs on chromosomes of sample 48. Chromosome 7 and 21 missed or gained one chromatid, respectively. Each chromosome is divided into color-coded bands. The short horizontal lines in different colors on the left side of the chromosome represent the SV composition, including deletion, insertion, duplication, translocation, and inversion, respectively, while those on the right side of the chromosome represent gain or loss of CNV in different bands.

**Figure 2 cancers-15-00035-f002:**
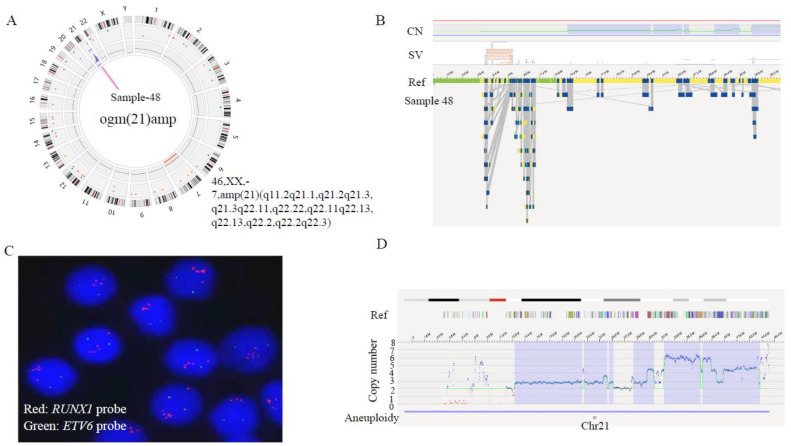
Identification of chromothripsis structures of chromosome 21 in sample 48 by OGM. (**A**) The circus plot illustrated the shattering of chromosome 21, leading to large-scale intra-chromosomal rearrangements. In addition, a monosomy of chromosome 7 was identified (indicated by the red line at the innermost layer of the circus). (**B**) Zoom-in to chromosome 21, showing the aberrant CNV profile(top) and maps of the rare variants (bottom). (**C**) Metaphase-FISH indicated 5 or more *RUNX1* signals (red). (**D**) OGM-CNV profile: the green line presents the copy number metrics of each fragment, which means different levels of gain at various breakpoints: 33.2–38.1 Mb fragment with CN = 6, 39.2–39.9 Mb fragment with CN = 4, 39.9–40.8 Mb fragment with CN = 3, 40.8–45.5 Mb fragment with CN = 4.

**Figure 3 cancers-15-00035-f003:**
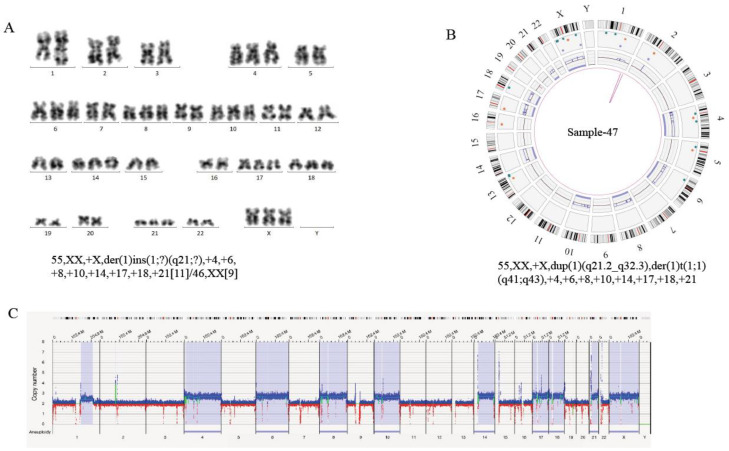
Production of an Accurate Cytogenomic Karyotype by Refining Cytogenetic Breakpoints and Resolving Unknown Cytogenetic Elements by OGM. (**A**) Standard G-banded karyotyping in sample 47 show the karyotype as 55, XX, +X, der(1)ins(1;?)(q21;?),+4,+6,+8,+10,+14,+17,+18,+21[11]/46,XX[9]. (**B**) Circus plot, showing a whole-genome view of a derivative chromosome 1 (pink lines) and copy number profiles (inner circle blue boxes indicate gains and red boxes indicate deletions). Thus, OGM identified additional intra-chromosomal translocation in chromosome 1 and refined the karyotype as 55, XX, +X, dup(1)(q21.2q24.3-q24.3q32.3), der(1) t(1;1)(q41;q43), +4,+6,+8,+10,+14,+17,+18,+21. (**C**) Whole-genome CNV profiles, generated with Bionano Access, indicates gains of chromosome 4, 6, 8, 10, 14, 17, 18, 21, 23. Some regions of chromosome 1 were amplified, and the *y* axis shows copy numbers ranging from 0–8 for each chromosome. The *x* axis shows increased copy numbers in blue and decreased copy numbers in red. The light blue region, highlighted for the above chromosomes, indicates a significant difference from the baseline, thus flagging a gain in those chromosomes. All chromosomes, except the sex chromosomes (Y)×0 and chromosomes (4, 6, 8, 10, 14, 17, 18, 21, 23)×3, are present in 2 copies.

**Figure 4 cancers-15-00035-f004:**
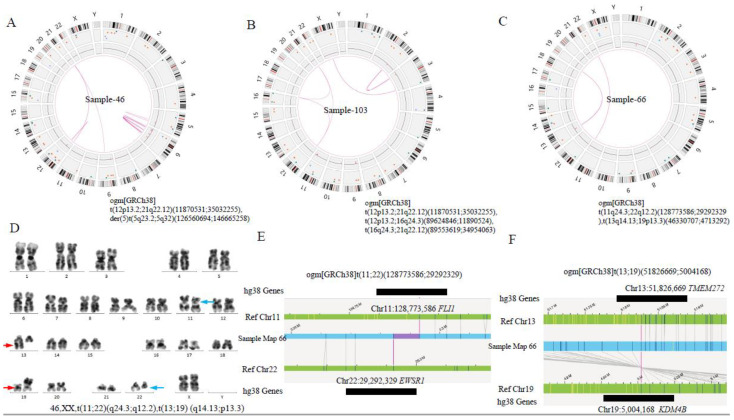
Identification of additional SV findings by OGM. (**A**) Circus plot showed a translocation between chromosomes 12 and 21 and the internal rupture of chr5, resulting in the formation of a derivative chromosome in case 46. (**B**) Circus plot showed three-way translocations between chr12 and 21, chr12 and 16, and chr16 and 21, respectively, in case 103. (**C**) Circus plot showed two translocations between chr11 and 22, chr13 and 19 in case 66, respectively. (**D**) Standard G-banded karyotyping showed t (11;22) (q24.3; q12.2) (blue arrow) and t(13;19) (q14.13;q13.3) (red arrow). (**E**,**F**) Linear genome browser representation of known *FLI1*::*EWSR1* fusion and a novel *TMEM272*::*KDM4B* fusion in case 66. Green represents GRCh38 reference chromosomes with OGM label patterns. Light blue represents assembled sample maps with label patters. Grey lines represent label alignments between two maps. Purple represents translocation breakpoints. Black lines represent the genes in this region.

**Figure 5 cancers-15-00035-f005:**
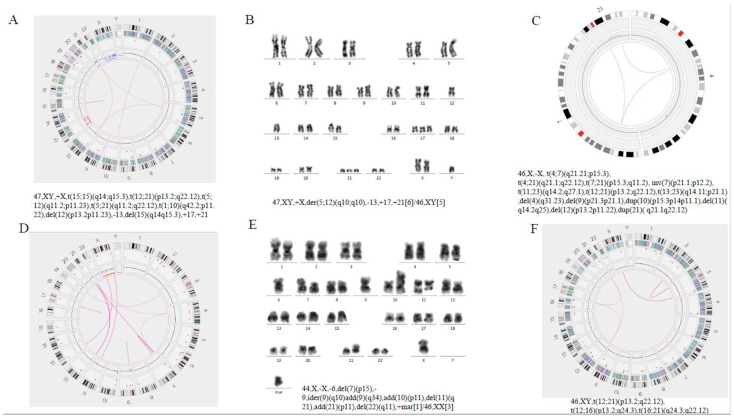
(**A**,**B**) Sample 114: A G-banded karyotype was performed and initially reported as the derivative chromosomes 5 and 12, add (17), add (21), add(X), del (13) in 6 metaphases with five metaphases with a normal karyotype. Optical genome mapping data confirm all known aberrations and show the presence of additional translocations t(15;15)(q14;q15.3), t(5;12)(q11.2;p11.23), t(5;21)(q11.2;q22.12), t(1;10)(q42.2;p11. 22). (**C**–**E**) Sample 83: Optical genome mapping data showing a three-way translocation 46,XX,t(4;7;21)(q21.21;p15.3;q11.2); multiple deletions (4,9,11,12,X) and duplications (10,21) on chromosome segments were also shown, as well as translocation between multiple chromosomes. Karyotype did not indicate multiple translocations between chromosomes. (**F**) Sample 103: Optical genome mapping confirm a three-way translocation on chromosomes 12, 16 and 21, while the G-banded karyotype (not represented in this plot) was normal.

**Figure 6 cancers-15-00035-f006:**
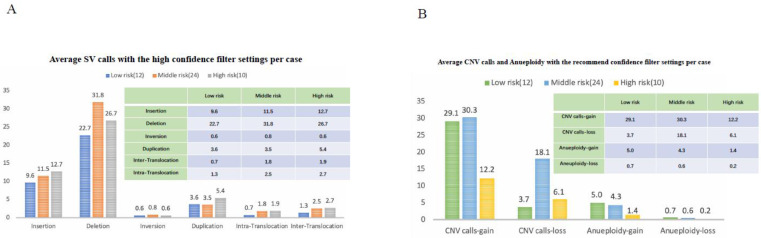
Chromosomal aberrations in 46 samples, analyzed by rare variant analysis using DLS marker. (**A**) Average SV with high filter settings per case. Different color bars represent different risk groups. The *x* axis showed different SV types, including insertion, missing, inversion, repetition, and translocation. The *y* axis showed the number of cases. (**B**) Average CNV and aneuploidy, with the recommended filter settings per case. The *x* axis showed the gain and loss of copy numbers and aneuploidy, while the *y* axis corresponded to the number of cases.

**Figure 7 cancers-15-00035-f007:**
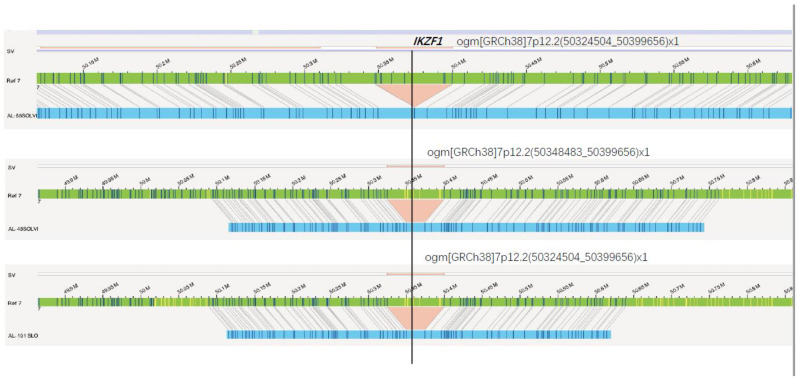
*IKZF1* deletion in 7p12.2 detected by OGM. The green bar indicates the reference map, patient maps (sample 48, 58, 101) were displayed in blue, and deletions were marked in red.

**Figure 8 cancers-15-00035-f008:**
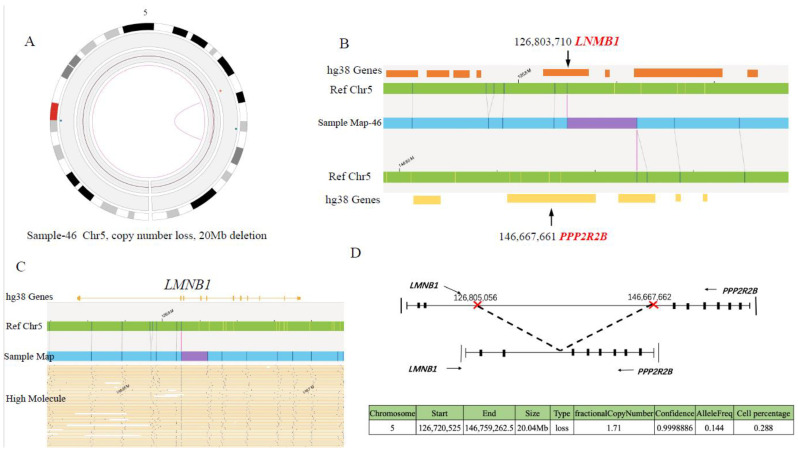
*LMNB1-PPP2R2B* fusion gene identified by OGM. (**A**) Zoom-in circus plot of chromosome 5 only, showing the fusion called by the SV and CNV tool. (**B**) Chromosome map view, navigated from circus plot, supports the existence of the fusion. The lower track shows the fusion call from the map, specifying the deletion between chr5:126,803,710 and chr5:146,667,661 (size: 20 Mbp). (**C**) OGM provides direct evidence for *LMNB1*::*PPP2R2B* fusion: aberrant molecules supporting the translocation. (**D**) Diagram of *LMNB1*::*PPP2R2B* fusion pattern.

**Table 1 cancers-15-00035-t001:** SVs and CNVs calling in OGM.

	Sum in All Samples	Average per Sample
**SV calls using the recommend confidence filter settings**		
Insertion	30,895	671.6
Deletion	29,099	632.6
Inversion	3105	67.5
Duplication	8256	179.5
Translocation inter-chromosomal	72	1.6
Translocation intrachromosomal	104	2.3
Total	71,531	1555.1
**SV calls using the high confidence filter settings**		
Insertion	517	11.2
Deletion	1301	28.3
Inversion	33	0.7
Duplication	180	3.1
Translocation inter-chromosomal	70	1.5
Translocation intra-chromosomal	103	2.2
Total	2204	47.0
**CNV calls (non-masked only)**		
Gain (called duplication in the file)	1189	25.8
Loss (called deletion in the file)	403	8.7
Total	1592	34.5
**Aneuploidy (non-masked only)**		
Aneuploidy Gain	177	3.8
Aneuploidy Loss	23	0.5
Total	200	4.3

**Table 2 cancers-15-00035-t002:** Additional SVs and CNVs, Identified by OGM but not by Conventional Technologies.

Sample ID	Karyotype	FISH/PCR	OGM[GRCh38]	Karyotype Predicted by OGM	Overlapping Genes	Additional Findings
AL-41	46,XX[16]	*RUNX1,IGH* and *CRLF2* copy number gain	(4)×3,(5)×3,(6)×3,(9p24.3-9q11)(14566-43279152)×3,(9q21.11-9q34.3)(68310629-138334464)×3,(10)×3,(14q11.2-14q32.33)(19761872-105833056)×3,(17)×3,(18)×3, (21q11.2-21q22.3)(13120706-45845512)×4, (23)×3	**53,XX,+X,+4,+5,+6, dup(9)(p24.3q11), dup(9)(q21.11q34.3),+10, dup(14) (q11.2q32.33), +17,+18, dup(21)(q11.2q22.3)**	*RUNX1, IGH, CRLF2*	Hyperdiploid
AL-45	45,XX,del(9)(p13) [10]	*MEF2D*::*HNRNPUL1*, *CRLF2* copy number gain	t(1q22;19q13.2)(156553387;41431238), t(9p24.1;9p21.2)(5025357;27143532), (9p24.2-p24.1)(3552238-8731962)×1,(9p22.3-9p21.3)(15980358-23496538)×1,(9p21.2-9p21.1)(96703784-96724287)×1,(12p13.33-12p12.33)(14568-16540369)×1,(Xp11.23-Xp22.2)(4170788-56416870)×3	**46,XX,t(1;19)(q22;q13.2),der(9) t(9;9)(p24.1;p21.2),del(9)(p24.2p24.1),del(9)(p22.3p21.3), del(9)(p21.2p21.1), del(12)(p13.33p12.33),dup(X)(p11.23p22.2)**	*MEF2D*::*HNRNPUL1, CRLF2,***J*AK2::TEK***	t(9;9)(p24.1;p21.2) **J*AK2::TEK***
AL-46	46,XX[9]	*ETV6*::*RUNX1*	t(12p13.2;21q22.12)(11870531;35032255),der(5),t(5q23.2;5q32)(126560694;146665258)	46,XX,t(12;21)(p13.2;q22.12), **der(5),t(5;5)(q23.2q32)**	*ETV6*::*RUNX1* ***LMNB1::PPP2R2B***	t(5;5)(q23.2q32) ***LMNB1::PPP2R2B***
AL-66	46,XX,t(11;22)(q23;q11);t(13;19)(q14;p13)[10]		t(11q24.3;22q12.2)(128773586;29292329), t(13q14.13;19p13.3)(46330707;4713292)	46,XX,t(11;22)(q24.3;q12.2),t(13;19) (q14.13;p13.3)	*FLI1*::*EWSR1; TMEM272*::*KDM4B,**AL162377.3*****::*****KDM4B***	t(13;19) (q14.13;p13.3) ***AL162377.3*****::*****KDM4B***
AL-74	46,XY[9]	*ZNF384* copy number loss	(12p13.311-12p12.1)(6670124-25189935) ×1, der(12) t(12p13.2;12p12.1)(11736073; 25175777), der(12)t(12p13.31;12p13.2) (6686961;11758879)	46,XY,der(12)t(12;12)(p13.2;p12.1),**der(12)t(12;12)(p13.31;p13.2)**	* **ETV6** * **::** * **CASC1, ZNF384** * **::** * **ETV6, ** * *ZNF384*	t(12;12)(p13.2;p12.1) ***ETV6::CASC1*** t(12;12)(p13.31;p13.2) ***ZNF384::ETV6***
AL-77	46,XX[9]	*RUNX1* and *CRLF2* copy number gain	(4)×3,(5)×3,(6)×3,(10)×3,(14)×3,(18)×3,(21)×3,(22)×3,(23)×3,(1q21.3)(151751846-15379814)×3,1q23.1q43(156616741_240089410)×3, (12p11.22-12q24.33)(27982715_132081687)×3, 17q11.1q25.3(26692353_81102800)×3, t(5;5)(q21.1;q33.2)(99,141,474;155,917,152), t(5;5)(q23.1;q34)(118,853,883;168,502,903), t(11;11)(q14.3;q24.1)(91,671,788;122,517,878), t(11;11)(q14.1;q25)(84,542,296;132,777,096)	**54,XX, +4,+5,+6, +10,+14,+18,+21, +22, +X, dup(1q21.3),dup(1q23.1q43), dup(12)(p11.2q24.33),dup(17)(q11.1q25.3), der(5)t(5;5)(q23.1;q34), der(5)t(5;5)(q21.1;q33.2), der(11)t(11;11)(q14.3;q24.1), der(11)t(11;11)(q14.1;q25)**	*RUNX1, CRLF2, **DTWD2*** **::** * **RARS** *	Hyperdiploid; t(5;5)(q23.1;q34) ***DTWD2::RARS***
AL-80	46, XX[1]	*MEF2D,RUNX1,IGH,CRLF2 and ETV6 copy number gain*	(1q21.1-1q44)(144085065_248943333)×3,(3)×3,(5)×3,(6)×3,(8)×3, (10)×4,(11)×3,(12p12.3-12p11.1) (19087028-34717936)×3, (12p13.31p12.3) (6245718-15306812)×1, t(12;16)(p13.31;p13.13)(6,245,718;11,607,398) t(9;12)(p24.1;p12.3)(19,087,028;5,761,495), (14)×4,(16p13.13-16 p11.2)(11584915-31987698)×3,(17q12-17q25.3)(38731593_82564467)×3,t(17;17)(q21.33;q22)(51,672,152;59,147,333), (18)×4,(21)×4,(23)×4	**56,XX,+X,dup(1)(q21.1q44),+3,+5,+6,+8,+10,+11,dup(12)(p12.3p11.1),del(12)(p13.31p12.3),+14,dup(16)(p13.13p11.2),dup(17)(q12q25.3),+18,+21, t(12;16)(p13.31;p13.13)** **t(9;12)(p24.1;p12.3)** **t(17;17)(q21.33;q22)**	*RUNX1, IGH, CRLF2, ETV6 MEF2D*	Hyperdiploid;
AL-97	46,XX[20]	*RUNX1,IGH* and *CRLF2* copy number gain	(4)×3,(6)×3,(9)×3,(10)×3,(14)×3,(18)×3,(21)×3,(X)×3	**54,XX,+X,+4,+6,+9+10,+14,+18,+21**	*RUNX1, IGH, CRLF2*	Hyperdiploid
AL-101	46,XY[8]	*RUNX1* copy number gain, *IKZF1* deletion	(7)×1,21q22.11(32926920-33533692)×3, 21q22.3(41288580-45259300)×3	**45,XY,−7,dup(21)(q22.11),dup(21)(q22.3)**	*RUNX1, IKZF1*	chr7 CN loss aenuploidy
AL-103	46,XY[20]	*ETV6*::*RUNX1*	t(12p13.2;21q22.12)(11870531;35032255),t(12p13.2;16q24.3)(89624846;11890524),t(16q24.3;21q22.12)(89553619;34954063)	46,XY,t(12;21)(p13.2;q22.12), **t(12;16)(p13.2;q24.3),t(16;21)(q24.3;q22.12)**	*ETV6*::*RUNX1, **ETV6*****::*****DPEP1, SPG7*****::*****RUNX1***	three-way trans; **t(12;16)*ETV6::DPEP1;*** **t(16;21)*SPG7::RUNX1***
AL-109	46,XX[9]	*RUNX1, IGH* and *CRLF2* copy number gain	(1q21.1-1q41)(145439805-215837313) ×3,(4)×3,(6)×3,(10)×3,(14)×3,(17)×3, (18)×3,(21)×4,(23)×4	**54,XX,+X,+X,dup(1)(q21.1q41),+4,+6,+14,+17,+18,+21**	*RUNX1, IGH, CRLF2*	Hyperdiploid

Note: The bold and underlined font indicates additional karyotype or fusion genes predicted by OGM vs. clinical karyotype findings; The predicted karyotype does not include clonal information.

**Table 3 cancers-15-00035-t003:** Refining Cytogenetic Breakpoints and Resolving Unknown Cytogenetic Elements by OGM.

Sample ID	Karyotype	FISH/PCR	OGM[GRCh38]	Karyotype Predicted by OGM	Overlapping Genes
AL-47	55,XX,+X,der(1) ins(1;?)(q21;?), +4,+6,+8,+10, +14,+17,+18,+21 [11]/46,XX[9]	*IGH* and *CRLF2* copy number gain	(1q21.2-1q32.3)(149910330-213101514)×3, der(1)t(1q41;1q43)(217001914;236988091), (4)×3,(6)×3,(8)×3,(10)×3,(14)×3,(17)×3,(18)×3, (21)×3,(23)×3	55,XX,+X,**dup(1)(q21.2 q32.3),der(1)t(1;1)(q41;q43),**+4,+6,+8,+10,+14,+17,+18,+21	*IGH, CRLF2, ESRRG*
AL-48	45 XX, −21, +mar[1]/45/idem,−7[14]/46, XX[5]	chr21(*iAMP21*),*P2RY8*::*CRLF2*,*IKZF1* deletion	(7)×1,(21q22.3)(45427332-46402888) ×1,(21q11.2-21q21.1)(14097084-25448211)×3,(21q21.2-21q21.3)(25448806-25975600)×4,(21q21.3-21q22.11)(26177986-31070035)×3,(21q22.11)(31077495-31942346)amp,(21q22.11-21q22.13)(33213701-37921102)amp,(21q22.13-21q22.2)(38134198-39259099)amp,(21q22.2)(39267796-39966121)amp,(21q22.2-21q22.3)(40804931-45514719)amp	46,XX,−7,**+21,amp(21)(q11.2q22.3)**	chr21(*iAMP21*),*IKZF1, APP, BRWD1,ERG,EST2,GET1*
AL-71	48~49, XX, +X, t(2;12)(p13;q24), −6,add(6)(q23), −7,−17,−20,+21, +3~5mar[7]/46,XX[13]	*RUNX1* and *CRLF2* copy number gain	(8)×3,(10)×3,(23)×3, (21)×3, 6q15q22.1(91357467-115474254)×1, 20q11.21q13.33(31182877-61256295)×1,17p13.3p11.2(1342670-20101698)×1, 9p24.3p13.1(14566-38890429)×1, 7p14.3q11.21(31801042-66549041)×1, 7q11.21q11.22(67337248-72514593)×1, t(2;12)(p11.2;q24.12)(88827954;111430007), t(6;7)(q22.1;q11.21)(66546520;116497165), der(7)t(7;7)(p14.3;q11.22)(31802046;67525321),t(7;10)(q11.21;q21.1)(51434361;67432607),t(6;10)(q15;q11.23)(51098504;91367054)	49, XX, +X,+8,+10,+21,−7,t(2;12)(p11.2;q24.12) **del(6)(q15q22.1),** **del(20)(q11.21q13.33),** **del(17)(p13.3p11.2),** **del(9)(p24.3p13.1),** **t(6;7)(q22.1;q11.21),** **der(7),t(7;7)(p14.3;q11.22),t(7;10)(q11.21;q21.1),** **t(6;10)(q15;q11.23)**	*AC244205.1*::*SH2B3,RUNX1, CRLF2*
AL-78	46, XX[16]	*ETV6*::*RUNX1*	t(12p13.2;21q22.12)(11870531;35032255), (12p13.33-12p12.3)(14568-17462436)×1,(22 q13.1-22q13.32)(38840431-48399879)×3, t(12;22)(p12.3;q13.1)(17474765;38809117) t(20;21)(p11.21;q22.12)(35029693;22329802)	46,XX,t(12;21)(p13.2;q22.12),del(12)(p13.33p12.3),dup(22)(q13.1q13.32), **t(12;22)(p12.3;q13.1),** **t(20;21)(p11.21;q22.12)**	*ETV6*::*RUNX1*
AL-85	46,XX,add(19)(p13)[2]/46,XX[5]	*TCF3*::*PBX1*	t(1p23.3;19q13.3)(164783197;1638016), (1q23.3-1q44)(164773702-248458732)×3,2p25.3(743869-1959263)×3,10q21.1(54833448-55584392)×1	46,XX,**t(1;19)(p23.3;q13.3),dup(1)(q23.3q44), dup(2)(p25.3),del(10)(q21.1)**	*TCF3*::*PBX1*
AL-107	55,XX,+X, der(1) ins(1;?)(q21;?), +4,+5,+6,+8,+10,+21, +21,+22[8]/55, idem, add(21) (q22)[2]/46,XX[10]	*RUNXL1* and *CRLF2* copy number gain	(1q21.1-1q32.3)(146397612-212907422) ×3,(4)×3,(5)×3,(6)×3,(8)×3,(10)×3,(21)×3, (22q11.21-22q13.1)(18746350-38096173)×3, (X)×3	**53,XX,+X,dup(1)(q21.1q32.3),**+4,+5,+6,+8,+10,+21,dup(22)(q11.21q13.1)	*RUNX1, CRLF2*

Note: The bold and underlined font indicates the Refining Cytogenetic elements by OGM vs. clinical karyotype findings; The predicted karyotype does not include clonal information.

**Table 4 cancers-15-00035-t004:** OGM identified all SVs detected by conventional technologies.

Sample ID	Karyotype	FISH/PCR	OGM[GRCh38]	Karyotype Predicted by OGM	Overlapping Genes
AL-42	/	*ETV6*::*RUNX1*	t(12p13.2;21q22.12)(11870531;35032255)	46,XY,t(12;21)(p13.2;q22.12)	*ETV6*::*RUNX1*
AL-49	46,XX,t(4;11)(q21;q23)[12]/46,XX[2]	*KMT2A*::*AF4*	t(4q21.3;11q23.3)(87082301;118477357)	46,XX, t(4;11) (q21.3;q23.3)	*KMT2A*::*AF4*
AL-58	46,XY,t(9;22)(q34;q11)[1]/45,sl,dic(7;9)(p12;p12)[17]/46,sdl,+der(22)t(9;22)[1]/46,XY[1]	*BCR*::*ABL1,IKZF1* deletion	t(9q34.12;22q11.23)(130864214;23203247),(2q32.2)(194855238-195411332)×1,(7p22.3-7p14.1)(205606-38213349) ×1, (7p12.2)(49320190-50264542) ×1,(7p12.3-p12.2)(48462939-49316285) ×3,(9p24.3-9p12)(585489-39591818) ×1,(19p13.2)(10015151-10983318) ×1	46,XY,t(9;22)(q34.12;q11.23),del(2)(2q32.3),del(7)(p22.3p14.1),del(7)(p12.2),dup(7)(p12.3p12.2,del(9)(p24.3p12),del(19)(p13.2)	*BCR*::*ABL1,IKZF1*
AL-65	46,XY[10]	*ETV6*::*RUNX1*	t(12;21)(p13.2;q22.12)(11870531;35032255), (21q11.1-21q22.3)(12983105-45259300)×3	46,XY, t(12;21)(p13.2;q22.12),dup(21)(q11.1q22.3)	*ETV6*::*RUNX1*
AL-82	46,XY[20]	*ETV6*::*RUNX1*	t(12p13.2;21q22.12)(11870531;35032255),(4q26-4q35.2)(117373259-190202564)×3,(12p13.33-12p13.2)(14568-11843683)×3	46,XY,t(12;21)(p13.2;q22.12),dup(4)(q26q35.2),dup(12)(p13.33p13.2)	*ETV6*::*RUNX1*
AL-83	44,X,-X,-6,del(7)(p15),-9,ider(9)(q10)add(9)(q34),add(10)(p11),del(11)(q21),add(21)(p11),del(22)(q11),+mar[1]/46,XX[3]	*ETV6*::*RUNX1*	t(4;7)(q21.21;p15.3)(78918774;24807169),t(4;21)(q21.1;q22.12)(75639432;34981504),t(7;21)(p15.3;q11.2)(24843132;14996044),inv(7)(p21.1;p12.2)(16927511;50407985),t(11;23)(q14.2;q27.1)(87167189;139837627),t(12;21)(p13.2;q22.12)(11870531;34981504),t(12;21)(p13.2;q21.1)(11881907;15483164),t(13;23)(q14.11;p21.1)(40417525;33567746) (X)×1,4q31.23(148400189-148979475)×1, 9p21.3p21.1(21148269-29886519)×1,der(9)t(9;9)(p21.3;p21.1)(21155066;29880139), 10p15.3p14p11.1(2660922-38780901)×3,11q14.2q25(87162780-135069565)×1,12p13.2p11.22(11809511-29745022)×1, 21q21.1q22.12(19535402-34928264)×3,	46,X,-X, t(4;7)(q21.21;p15.3), t(4;21)(q21.1;q22.12),t(7;21)(p15.3;q11.2), inv(7)(p21.1;p12.2), t(11;23)(q14.2;q27.1),t(12;21)(p13.2;q22.12),t(13;23)(q14.11;p21.1),del(4)(q31.23),del(9)(p21.3p21.1),dup(10)(p15.3p14p11.1),del(11)(q14.2q25),del(12)(p13.2p11.22),dup(21)(q21.1q22.12),	*ETV6*::*RUNX1* *BZW2*::*IKZF1* *OSBPL3*::*NRIP1* *PAQR3*::*OSBPL3* *OSBPL3*::*AF127577.4*
AL-84	47,XY,+X,t(9;11)(p22;q23)[16]/46,XY[4]	*KMT2A*::*MLLT3*	t(9p21.3;11q23.3)(20358621;118493942),(X)×3,22q11.22(22010337-22908320)×1, t(11;11)(q14.1;q21)(77603065;94621267)	47,XY,+X,t(9;11)(p21.3;q23.3),inv(11)(q14.1;q21), del(22)q11.22	*KMT2A*::*MLLT3*
AL-89	46,XY[20]	*IGH* copy number gain	(14q11.2-14q32.33)(22525374-104169671)×3,(21q11.1-21q22.3)(12406577-43289581)×3	46,XY,dup(14)(q11.2q13.1q32.33),dup(21)(q11.1q22.3)	*IGH*
AL-95	45,XX,−7, add(9)(p13)[10]/46,idem,+mar[4]/46,XX[6]	*TCF3*::*ZNF384; ZNF384* copy number gain	t(12p13.31;19p13.3)(1643841;6674678), (7)×1,(12p13.33-12p13.31)(377048-6670124) ×3	45,XX,−7,t(12;19)(p13.31;p13.3),dup(12)(p13.33p13.31)	*TCF3*::*ZNF384, ZNF384*
AL-98	56,XX,+X,+2,+4,+6,t(9;22)(q34;q11),+10,+15,+18,+21,der(22)t(9;22),mar[4]/55,idem,-15,add(12)(q24)[7]/46,XX[9]	*BCR*::*ABL1,RUNX1* copy number gain	t(9q34.12;22q11.23)(130732573;23244051),(2)×3,(4)×3,(6)×3,(10)×3,(12q24.21-1q24.33)(115355442-133263960)×1,(15)×3,(18)×3,(21)×3,(22q11.21-22q11.23)(18636137-23191585)×3,(X)×3	56,XX,+X,+2,+4,+6,t(9;22)(q34.12;q11.23),+10,del(12)(q24.21q24.33),+15,+18,+21,dup(22)(q11.21q11.23)	*BCR*::*ABL1,RUNX1*
AL-114	47,XY,+X,der(5;12)(q10;q10),-13,+17,+21[6]/46,XY[5]	*ETV6*::*RUNX1, CRLF2* copy number gain	der(15)t(15;15)(q14;q15.3)(38438035;43496447),t(12;21)(p13.2;q22.12)(11870531;34883313),t(5;12)(q11.2;p11.23)(27128273;58971017),t(5;21)(q11.2;q22.12)(34899705;58971017),t(1;10)(q42.2;p11.22)(31401031;233989381) (X)x2,(17)×3,(21)×3,(13)×1, 12p13.2p11.23(11845928-27123509)×1,15q14q15.3(38411329-43516043)×1,	47,XY,+X,t(15;15)(q14;q15.3),t(12;21)(p13.2;q22.12),t(5;12)(q11.2;p11.23), t(5;21)(q11.2;q22.12),t(1;10)(q42.2;p11.22),del(12)(p13.2p11.23),-13,del(15)(q14q15.3),+17,+21	*ETV6*::*RUNX1,CRLF2*

The predicted karyotype does not include clonal information.

**Table 5 cancers-15-00035-t005:** Result comparison between OGM and conventional technologies.

Methods	Aneuploidies Only	Translocation and/or Aneuploidies	Negative Karyotype	Total
Conventional technologies	18 *	11	17	46
OGM	22	11	13 *	46
OGM concordance	100%	100%	100%	

* Four (case 41, 97, 101, 109) with normal karyotype had additional SVs identified by OGM.

**Table 6 cancers-15-00035-t006:** Correlations of recurrent putative fusion genes with clinical features of B-ALL in children.

Putative Gene Fusion	Clinical Features	Spearman ρ	*p*
*BCR::ABL1*	WBC	0.297	0.045
*GPN3::FAM216A*	WBC	0.313	0.034
*AC026202.2::EDEM1*	WBC	0.305	0.039
*ARL8B::EDEM1*	WBC	0.305	0.039
*MTAP::CDKN2B-AS1*	WBC	0.322	0.029
*GRAPL::AC106017.1*	age	0.378	0.01
*GRAPL::KYNUP3*	age	0.378	0.01
*ETV6::AP000331.1*	d15 MRD	−0.307	0.04
*AC141586.1::KCTD5*	d33 MRD	0.405	0.006
*ATP10A::AC016266.1*	d33 MRD	0.329	0.029
*CALCOCO2::SUMO2P17*	d33 MRD	0.350	0.020
*MIR4435.2HG::AC017002.5*	d33 MRD	0.298	0.049
*AL034430.1::SLX4IP*	d78 MRD	0.318	0.038
*MKKS::SLX4IP*	d78 MRD	0.318	0.038
*AC141586.1::KCTD5*	Risk stratification	0.318	0.033
*ATP10A::AC016266.1*	Risk stratification	0.318	0.033
*CALCOCO2::SUMO2P17*	Risk stratification	0.318	0.033
*PDCD6IPP1::AC138649.1*	Risk stratification	0.305	0.041
*AC133919.2::LINC02193*	Percentage of blasts in PB at diagnosis	−0.388	0.008
*FAM157C::LINC02193*	Percentage of blasts in PB at diagnosis	−0.388	0.008
*GPN3::FAM216A*	Percentage of blasts in PB at diagnosis	0.297	0.045

PB: peripheral blood.

**Table 7 cancers-15-00035-t007:** Correlation of recurrent single-gene aberrations annotated to cosmic database with clinical features of B-ALL in children.

Genes	Gene Alteration Type	Clinical Features	Spearman ρ	*p*
*BCR*	Inter-trans	WBC	0.297	0.045
*ABL2*	Inter-trans	WBC	0.297	0.045
*TCF3*	Inter-trans	WBC	0.321	0.030
*IKZF1*	Intra-trans, del	age	0.361	0.014
*ERG*	dup	d15MRD	0.361	0.035
*NF1*	del	d15MRD	0.302	0.044
*CREBBP*	del	d33MRD	0.340	0.024
*ERG*	dup	d33MRD	0.372	0.013
*IKZF1*	Intra-trans, del	d33MRD	0.394	0.008
*NF1*	del	d33MRD	0.283	0.042
*SH2B3*	Inter-trans, del	d33MRD	0.329	0.029
*BTG1*	del	d78MRD	0.383	0.011
*CREBBP*	del	d78MRD	0.488	0.001
*KMT2A*	Inter-trans, del	d78MRD	0.363	0.017
*PIK3CA*	del	d78MRD	0.463	0.002
*CREBBP*	del	Risk stratification	0.318	0.033
*ERG*	dup	Risk stratification	0.318	0.033
*KMT2A*	Inter-trans, del	Risk stratification	0.394	0.007
*SH2B3*	Inter-trans, del	Risk stratification	0.318	0.033

Note: Inter-trans: inter-translocation; del: deletion; dup: duplication.

## Data Availability

Please contact author for data requests.

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
