# Peer review of "Optical Genome Mapping for Comprehensive Assessment of Chromosomal Aberrations and Discovery of New Fusion Genes in Pediatric B-Acute Lymphoblastic Leukemia"

_cancers, 2022, doi:10.3390/cancers15010035_

Round 1

Reviewer 1 Report

The manuscript by Huixia Gao and co-authors adds to the recent studies evaluating the usefulness of OGM technology in childhood B-ALL in children. It includes a relatively large cohort of children and definitely contains some interesting data and merits publication.

Nevertheless, the manuscript could be improved and below are my suggestions.

Abstract

Line 16&17 “Several novel fusion genes and single 16 gene mutations associated with definite or potential pathologic 17 significance that hadn’t been detected by traditional methods 18 were also identified. The word mutation should be replaced by alterations since OGM does not allow detection of point mutations and this is misleading.

Methods

Unusual high success rate of UHMW DNA isolation and processing for OGM analysis since all samples passed the quality criteria. It would be useful for readers if the authors include more details about the used bone marrow samples, such as cell count per ml, how many cells were used for extraction, were only fresh samples used or were also frozen samples included, minimum DNA yield that was accepted to proceed).  

Table 2,3 & 4

From the data provided in the tables G-band analysis performed by the authors seems capable to reveal the recurrent translocation t(12;21) resulting in the fusion ETV6-RUNX1. This translocation is considered cryptic (not detectable) by conventional cytogenetic analysis with the need of identification by FISH or PCR testing. The authors should clarify how their analysis was able to detect this translocation in sample 46, 51, 65, 78 and 82 or explain f they adjusted the nomenclature based on the FISH or PCR findings. In that case the ISCN nomenclature should including of the other method used.

Figure 2

The complex pattern of chromosome 21 revealed by OGM in the case with iAMP21 is explained as a chromothripsis. However iAMP21 is described as a complex genetic mechanism so called breakage-fusion-bridge (BFB) which is expected to show various breakpoints and different levels of gains ( ref. Sinclair PB et al. Human Molecular Genetics, Volume 20, Issue 13, 1 July 2011, Pages 2591–2602. It would be very helpful to add to the figure a chromosome 21 CNV profile to compare the complex pattern of chromosome 21 with illustrations from this alterations reported in the literature. Comparing the copy number profiles from OGM and from micro-array or WGS would better illustrate whether the complexity observed by OGM is the BFB mechanism and in addition a chromothripsis of 21.

2.3 identification of novel chromosome alterations or gene fusions

Confirmation of these finding with conventional methods such as FISH on metaphase chromosomes or long read NGS should be added to the study since this is a validation of a novel technology and to critically investigate the presence of false positive findings.. Illustration of the detected three-way translocation identified by OGM confirmed by other methods should be added in this chapter as well as to the figures.

3.2  correlation of recurrent SVs with common clinical characteristics in B-ALL

The authors should include correlation of recurrent CNV which are currently used in various clinical trials to stratify patients such as IKZFplus profile by Stanulla M. et al J Clin Oncol 2018 Apr 20;36(12):1240-1249, Steeghs et al  Sci Rep. 2019 Mar 15;9(1):4634, Hamadeh L et al Blood Adv 2019 Jan 22;3(2):148-157

Table 7 should indicate which type of single gene alteration ( gain, loss, insertion?) since this will obviously have highly variable molecular impact on the protein

Discussion.

Page 22 Line 28: The fact that OGM missed the P2RY8-CRLF2 fusion which is due to a deletion in the PAR region that seems poorly covered by OGM is mentioned briefly but the authors should elaborate on the fact that this alterations has been repeatedly reported to have high clinical significance is included in the stratification of patient in clinical trials since it enables treatment by TKI inhibitors.

Page 23 line 9. “Our results show that OGM can detect all clinical relevant SV …. Etc. Is exaggerated since P2RY8-CRLF2 is not detected by OGM according to this study.

General remarks.

-       ETV6-RUNX1 and TEL-AML1 fusions were referred to randomly in the text. This is confusing and only gene names according to HUGO gene nomenclature should be used.

-       The manuscript would benefit to include in the discussion if patients from the 46 individuals included in this study would have been stratified differently or if OGM analysis it would have had an impact on patient management if the analysis would have been performed at diagnosis in a clinical setting.   

Author Response

Manuscript ID number:

Cancers-1961289

Title of paper:

Optical genome mapping for comprehensive assessment of chromosomal aberrations and discovery of new fusion genes in pediatric B-Acute Lymphoblastic Leukemia

Dear Ms. Milica Grujic:

On behalf of my co-authors, we are very grateful for revising our manuscript. Meanwhile, we appreciate the editor and reviewer's positive and constructive comments and suggestions on our manuscript. We have studied the reviewer's comments carefully and have revised the paper. We have tried our best to revise our manuscript according to the comments. Please find the revised version attached, which we would like to submit for your kind consideration. Revised portions are marked in red on the paper. We want to express our great appreciation to you and the reviewers for your comments on our paper. The modifications are listed below:

Reviewer: 1

Abstract

Line 16&17 “Several novel fusion genes and single gene mutations associated with definite or potential pathologic significance that hadn’t been detected by traditional methods were also identified. The word mutation should be replaced by alterations since OGM does not allow detection of point mutations and this is misleading.

Reply: Thank you for your detailed proposal. “gene mutations” has been corrected as your precise modification proposal (Please see Page 2 Line 17).

Methods 

Unusual high success rate of UHMW DNA isolation and processing for OGM analysis since all samples passed the quality criteria. It would be useful for readers if the authors include more details about the used bone marrow samples, such as cell count per ml, how many cells were used for extraction, were only fresh samples used or were also frozen samples included, minimum DNA yield that was accepted to proceed).  

Reply: Thank you for your comment. Details have been added in Page 4 Line 1-6,14-15). In summary,UHMW gDNA extraction from frozen bone marrow aspirates (BMA) was performed, for each sample,1 mL BMA as start material, and a minimum of 1.5 million white blood cells was used to purify UHMW DNA. Finally, a total of 750 ng UHMW gDNA was labeled specifically according to the manufacturer’s guidelines.

Table 2,3 & 4

From the data provided in the tables G-band analysis performed by the authors seems capable to reveal the recurrent translocation t(12;21) resulting in the fusion ETV6-RUNX1. This translocation is considered cryptic (not detectable) by conventional cytogenetic analysis with the need of identification by FISH or PCR testing. The authors should clarify how their analysis was able to detect this translocation in sample 46, 51, 65, 78 and 82 or explain f they adjusted the nomenclature based on the FISH or PCR findings. In that case the ISCN nomenclature should including of the other method used. 

Reply: Thank you for pointing out our mistakes. As you have mentioned, the recurrent translocation t(12;21) resulting in the fusion ETV6-RUNX1 is cryptic to karyotyping, while FISH or PCR detected ETV6-RUNX1 fusion. We mistakenly added t (12; 21) into the results of karyotyping in the manuscript. With your reminder, we have corrected the karyotyping results of samples 46,51,65,78 and 82 in Tables 2-4. Thank you again for your professional correction.

Figure 2 

The complex pattern of chromosome 21 revealed by OGM in the case with iAMP21 is explained as a chromothripsis. However, iAMP21 is described as a complex genetic mechanism so called breakage-fusion-bridge (BFB) which is expected to show various breakpoints and different levels of gains (ref. Sinclair PB et al. Human Molecular Genetics, Volume 20, Issue 13, 1 July 2011, Pages 2591–2602. It would be very helpful to add to the figure a chromosome 21 CNV profile to compare the complex pattern of chromosome 21 with illustrations from this alteration reported in the literature. Comparing the copy number profiles from OGM and from micro-array or WGS would better illustrate whether the complexity observed by OGM is the BFB mechanism and in addition a chromothripsis of 21. 

Reply: Thank you for your detailed proposal. Based on your professional guidance, we have added the OGM-CNV profile (Figure 2D) with new figure legend (the green line presents the copy number metrics of each fragment, which means different levels of gain at various breakpoints: 33.2-38.1Mb fragment with CN=6, 39.2-39.9Mb fragment with CN=4, 39.9-40.8Mb fragment with CN=3, 40.8-45.5Mb fragment with CN=4). These patterns were consistent with the classical breakage – fusion – bridge (BFB) cycle of oncogene amplification. At the same time, we also observed some secondary genetic changes in this sample, including -7, P2RY8-CRLF2 and IKZF1deletion, etc. By comparing copy number profiles from the OGM with those from previous microarray studies, the altered complexity observed by the OGM brings an intuitive visual illustration of the BFB mechanism. (Please see Page 11 Line 9-12, Page 12 Line 4-18)

2.3 identification of novel chromosome alterations or gene fusions

Confirmation of these finding with conventional methods such as FISH on metaphase chromosomes or long read NGS should be added to the study since this is a validation of a novel technology and to critically investigate the presence of false positive findings. Illustration of the detected three-way translocation identified by OGM confirmed by other methods should be added in this chapter as well as to the figures.

Reply: Thank you for your detailed proposal. For the novel fusion genes detected by OGM, it really needs to be further verified by traditional methods. The two fusion events detected by OGM in case 66 were also detected by G-band karyotyping (Figure 4C-4D), among which t(11; 22) leading to FLI1-EWSR1 fusion has been reported by others previously, whereas TMEM272-KDM4B caused by t(13; 19) has not been reported yet. In the section "4.1NGS validation of SVs detected by OGM", we described the verification of this new fusion gene.

In this study, OGM identified balanced cryptic complex karyotypes leading to three-way translocation (the OGM-Circos plot shows a similar triangular structure, as shown in Figure 5), whereas the G-banding method does not reveal these cryptic complex anomalies. As you pointed out, we do need to verify these novel three-way translocations by conventional methods. Unfortunately, due to running out of previously clinical samples collected, no raw BMA nor DNA is available for verification now. We will focus on these rare three-way translocations in future studies and further verify their clinical significance. We have added these explanations in our manuscript. Please see Page 14 Line 13-17.

3.2  correlation of recurrent SVs with common clinical characteristics in B-ALL

The authors should include correlation of recurrent CNV which are currently used in various clinical trials to stratify patients such as IKZFplus profile by Stanulla M. et al J Clin Oncol 2018 Apr 20;36(12):1240-1249, Steeghs et al  Sci Rep. 2019 Mar 15;9(1):4634, Hamadeh L et al Blood Adv 2019 Jan 22;3(2):148-157

Reply: Thank you for pointing out our deficiency in writing. Here we slightly adjusted the order of expression and added citations to illustrate the correlation of recurrent CNV with clinical characteristics. (Please see Page 16 Line 32-33 and Page 17 Line 3-14)

Table 7 should indicate which type of single gene alteration (gain, loss, insertion?) since this will obviously have highly variable molecular impact on the protein.

Reply: Thank you for your detailed proposal. Based on your professional guidance, we have added the mutation types of single gene changes in Table 7.

Discussion.

Page 22 Line 28: The fact that OGM missed the P2RY8-CRLF2 fusion which is due to a deletion in the PAR region that seems poorly covered by OGM is mentioned briefly but the authors should elaborate on the fact that this alterations has been repeatedly reported to have high clinical significance is included in the stratification of patient in clinical trials since it enables treatment by TKI inhibitors. 

Reply: Thank you for your professional comment. After your reminder, we realized the special significance of P2RY8-CRLF2 fusion and made a supplementary description. We also highlight that OGM may have the potential to become a routine tool for hematology malignancies, but its technical limitations still need to be further addressed, especially in cases where complementary tools are needed, such as anomalies involving centromeres and PARs (Please see Page 22 Line 29-41).

Page 23 line 9. “Our results show that OGM can detect all clinical relevant SV …. Etc. Is exaggerated since P2RY8-CRLF2 is not detected by OGM according to this study.

Reply: Thank you for your comment. Given the fact that OGM missed the P2RY8-CRLF2 fusion which is due to a deletion in the PAR region that seems poorly covered by OGM, we realized that our description of our goals was indeed overreaching, and we’ve tweaked the expression a bit to avoid overstating the truth (Please see Page 23 Line 22-24).

General remarks

-       ETV6-RUNX1 and TEL-AML1 fusions were referred to randomly in the text. This is confusing and only gene names according to HUGO gene nomenclature should be used. 

Reply: Thank you for your professional and accurate review suggestions. After your reminder, we realized that our description was not rigorous enough, and we have replaced the description of TEL-AML1 in the text with the professional name (ETV6-RUNX1) certified by HGNC. We'll pay attention to these details in the future. (Please see Page 25 Line 4-6)

-       The manuscript would benefit to include in the discussion if patients from the 46 individuals included in this study would have been stratified differently or if OGM analysis it would have had an impact on patient management if the analysis would have been performed at diagnosis in a clinical setting.  

Reply: Thank you for your detailed proposal and pointing out our deficiency in writing. Most of the known fusion genes and important single gene alterations involved in the current stratification standard detected by traditional methods also have been detected by OGM. From that aspect, OGM is not having additional impact on stratification nor patient’s management comparing with the portfolio of conventional tests limited to the 46 cases included in this study. However, OGM detected additional clinically relevant SVs and fusions, which need further verification by larger cohort study by us and by other scientists. Those may be proved to have impact on clinical stratification or patient management in future. We have added these descriptions in our manuscript. Please see Page 23 Line 39-42 and Page 23 Line 44-48.

After careful consideration and full discussion, we have responded to your comments. Frankly speaking, your accurate and professional comments made us aware of our problems in expression and lack of critical thinking. After modification according to your suggestions, our work became more valuable. Finally, thanks again for your help in our research work.

Reviewer 2 Report

In the manuscript “Optical genome mapping for comprehensive assessment of chromosomal aberrations and discovery of new fusion genes in pediatric B-Acute Lymphoblastic Leukemia” Gao et al., examined chromosomal aberration of 46 pediatric B-ALL patients through Optical Genomic Mapping (OGM) and showed that OGM is highly effective in identifying chromosomal aberrations and has important implications for risk stratification of ALL and the pathogenesis of leukemia. Though the sample size is small and from a single center, the overall study was impressive.  

However, my comments are as follows:

1.    The newly identified genomic variations and fusion genes should be submitted to ClinVar-NCBI https://www.ncbi.nlm.nih.gov/clinvar/  or GenBank and provide the accession numbers in the manuscript.

2.    In Figures 2B and D, it is difficult to read the letters. Higher-resolution images need to be provided.

3.    Similarly, Figures 3B, Figure 4E, and F, and Figures 8A, B, and C need to include the higher-resolution image.

4.    In scheme 3 D authors try to show the LMNB1-1 PPP2R2B fusion in 396 ALL samples through an agarose gel. However, I could not see clear bands at 220bp. Where the authors sequenced from this PCR product. So, the authors need to represent another gel with a clear PCR band.

In figure scheme 3 C it is difficult to find the variations, that need to improve the higher resolution.  I also recommend submitting the LMNB1-1 PPP2R2B fusion sequence to GenBank or ClinVar to confirm the same.

5.    Supplementary Figure A1 figure legend is missing. 

Author Response

Manuscript ID number:

Cancers-1961289

Title of paper:

Optical genome mapping for comprehensive assessment of chromosomal aberrations and discovery of new fusion genes in pediatric B-Acute Lymphoblastic Leukemia

Dear Ms. Milica Grujic:

On behalf of my co-authors, we are very grateful for revising our manuscript. Meanwhile, we appreciate the editor and reviewer's positive and constructive comments and suggestions on our manuscript. We have studied the reviewer's comments carefully and have revised the paper. We have tried our best to revise our manuscript according to the comments. Please find the revised version attached, which we would like to submit for your kind consideration. Revised portions are marked in red on the paper. We want to express our great appreciation to you and the reviewers for your comments on our paper. The modifications are listed below:

Reviewer: 2

  1. The newly identified genomic variations and fusion genes should be submitted to ClinVar-NCBI https://www.ncbi.nlm.nih.gov/clinvar/ or GenBank and provide the accession numbers in the manuscript. 

Reply: Thank you for your comment. Due to the Regulation on the Management of Human Genetic Resources of China, it takes some time to get the approval to submit the corresponding data requested. Once we get the approval, I will immediately upload newly identified genomic variations and fusion genes found by OGM to the ClinVar-NCBI website for verification according to your guidance.

  1. In Figures 2B and D, it is difficult to read the letters. Higher-resolution images need to be provided. 

Reply: Thank you for your comment. Figures 2B and D have been updated. In order to illustrate the copy number change of RUNX1 more clearly, we have adjusted the CNV profile diagram.

  1. Similarly, Figures 3B, Figure 4E, and F, and Figures 8A, B, and C need to include the higher-resolution image. 

Reply: Thank you for your comment. Figures 3B, Figure 4E, and F, and Figures 8A, B, and C have been updated.

  1. In scheme 3 D authors try to show the LMNB1-1 PPP2R2B fusion in 396 ALL samples through an agarose gel. However, I could not see clear bands at 220bp. Where the authors sequenced from this PCR product. So, the authors need to represent another gel with a clear PCR band. 

In figure scheme 3 C it is difficult to find the variations, that need to improve the higher resolution.  I also recommend submitting the LMNB1-1 PPP2R2B fusion sequence to GenBank or ClinVar to confirm the same. 

Reply: Thank you for your professional guidance and suggestions. We have improved this figure to show the band in agarose gel clearly. Figure scheme 3 C showing the variation more clearly has been updated and the “G” indicated by the red arrow is the break-fusion point site. Meanwhile, same situation as Question 1 and we will submit the LMNB1-PPP2R2B fusion sequence to ClinVar for verification once we get the approval to upload data.

  1. Supplementary Figure A1 figure legend is missing. 

Reply: Thank you for your comment. Figure legend has been added as below: Supplementary Figure 1(A)WGS sequencing maps showed a deletion of about 0.2Mbp in chr13 (19,761,221-20,037,079) which leads to formation of PSPC1-ZMYM2 fusion. (B) WGS confirmed the existence of SH2B3-ATXN3 fusion caused by a deletion of approximately 0.02Mbp in chromosome 12. (C) Diagram of CWH43 and TPTE fusion pattern.

After careful consideration and full discussion, we have responded to your comments. Frankly speaking, your accurate and professional comments made us aware of our problems with chart production. After modification according to your suggestions, our work became clearer. Finally, thanks again for your help in our research work.

Round 2

Reviewer 1 Report

Propositions for improvement and comments  have been adequately adressed 

Author Response

Thank you.

Reviewer 2 Report

Some of the updated figures in the manuscript still NOT convincing including, PCR gel images. It is very important because the whole experiment is based on sequencing. However, the PCR bands are unclear which makes the PCR study dubious.

Figures Scheme 2 B and C images are still not clear (I could NOT read the letters in the figure), please include the higher-resolution images.

Example: At the red arrow it is difficult to find which codon is changed? Clarification is needed.

Similarly, I could NOT read anything at the bottom of panel C, it should be replaced with the higher-resolution figure.

Figure scheme 3 C also, needs to be replaced with a high-resolution figure with readable letters. In the current form, it is difficult to find which codon is changed at the red arrow.   

Authors claim that “We have improved this figure to show the band in agarose gel clearly”. However, the authors kept the same gel. I did NOT see the proper band in the left panel. In Figure legend “D) Agarose gel electrophoresis showed another case with LMNB1-1 PPP2R2B fusion in 396 ALL samples”. They mentioned “LMNB1-1 PPP2R2B fusion was seen in 396 ALL samples” So please provide another gel image with clear bands.  

Author Response

Some of the updated figures in the manuscript still NOT convincing including, PCR gel images. It is very important because the whole experiment is based on sequencing. However, the PCR bands are unclear which makes the PCR study dubious.  

Q1: Figures Scheme 2 B and C images are still not clear (I could NOT read the letters in the figure), please include the higher-resolution images. Example: At the red arrow it is difficult to find which codon is changed? Clarification is needed

Reply: Thank you for your professional guidance and detailed suggestions.

The details of the fusion junction indicated by the red arrow have been magnified according to your suggestion. Please see Supplementary Figure 2B.

Q2: Similarly, I could NOT read anything at the bottom of panel C, it should be replaced with the higher-resolution figure.

Reply: Thank you for pointing out the imperfections in our drawing.

As you have mentioned, higher-resolution figure has been replaced as follows. Please see Supplementary Figure 2C.

Q3: Figure scheme 3 C also, needs to be replaced with a high-resolution figure with readable letters. In the current form, it is difficult to find which codon is changed at the red arrow.   

Reply: We are very sorry for the inconvenience caused by the same problem again. The detail diagram has been replaced as follows. Please see Supplementary Figure 3C.

Q4: Authors claim that “We have improved this figure to show the band in agarose gel clearly”. However, the authors kept the same gel. I did NOT see the proper band in the left panel. In Figure legend “D) Agarose gel electrophoresis showed another case with LMNB1-1 PPP2R2B fusion in 396 ALL samples”. They mentioned “LMNB1-1 PPP2R2B fusion was seen in 396 ALL samples” So please provide another gel image with clear bands. 

Reply: Thank you for your professional guidance and suggestions. We are sorry that we cannot repeat the experiment due to the insufficient initial samples. We have tried our best to adjust the background to show the band in the left figure as follows. Please see Supplementary Figure 3D.

After careful consideration and full discussion, we have responded to your comments. Frankly speaking, your accurate and professional comments made us aware of our problems with chart production. After modification according to your suggestions, our work became clearer. Finally, thanks again for your help in our research work.

Round 3

Reviewer 2 Report

Look like the authors addressed all the comments. I recommend the manuscript to accept in the Cancers.    

Author Response

Thank you.